# How survey results are reported in the media: A framework on selection mechanisms and a pilot study on reporting practice

**Jenny Bethäuser**[1]*, **Natalja Menold**[2], **Peter Winker**[1]

**1** Faculty of Economics and Business Studies, Justus Liebig University Giessen, Giessen, Germany,
**2** Institute of Sociology, Dresden University of Technology, Dresden, Germany

* jenny.bethaeuser@wirtschaft.uni-giessen.de

**Data availability statement:** The data underlying the empirical results presented in the

## Abstract

Survey results are frequently reported in the media without sufficient background information to assess their methodological quality. This gap may arise from various factors, including a lack of statistical literacy among those involved in generating or disseminating results, time and space constraints in journalistic contexts, or the perception that methodological details may overwhelm or confuse audiences. In addition, economic (dis)incentives in the media landscape can affect how journalists decide what to include or leave out when reporting on survey results. We propose a conceptual framework for these factors and their interactions and outline their expected implications for the communication of survey-based findings in public discourse. This framework serves as the foundation for a subsequent empirical pilot study focused on reporting in DER SPIEGEL and BILD. We collected articles published in the second half of 2023 that include the key word 'survey'. We performed a manual content analysis to determine the background information provided on the survey methodology when referring to the survey results. We found that the source of the survey is the most frequent and often the only information on survey background provided. The proposed framework could serve as the foundation for a subsequent broader empirical analysis aimed at identifying the main drivers that influence media reporting on survey statistics and guiding improvements in statistical literacy and reporting practices.

## Introduction

The headlines in the media landscape serve as powerful tools to captivate readers, pique their curiosity, and encourage them to delve deeper into the subject matter. Regardless of whether one navigates the world of traditional print media, such as newspapers and magazines, or ventures into the realm of online publications, headlines play a significant role in shaping readers' perceptions and interests. Frequently, the results of surveys emerge as compelling hooks or titles, further enhancing the appeal of the news articles or reports they accompany.

study are available from
https://doi.org/10.22029/jlupub-19855.

**Funding:** The author(s) received no specific funding for this work.

**Competing interests:** The authors have declared that no competing interests exist.

When it comes to socioeconomic issues of current interest, citing survey results seems to provide some sort of scientific legitimacy, even when the surveys themselves may not follow rigorous scientific standards, but are instead commissioned by the media outlets themselves.

As an example illustrating the use of study findings as evidence for factual assertions, we consider a recent communication by Plan International (www.plan.de), a non-governmental organization dedicated to development cooperation and humanitarian aid with a focus on children. On its website, Plan International captured the reader's attention with the headline that one-third of surveyed men have resorted to physical violence against women to instill respect and perceived occasional physical altercations with their partners as acceptable [1]. SPIEGEL+ reported on this survey on 11 June 2023 under the headline "One in three young men supports violence against women" in its Panorama section and followed up the next day in the Science section with the headline "One in three men finds violence against women acceptable, or not?" which included some critical comments on the methodology and featured an expert interview published on SPIEGEL online on 16 June 2023. The nationwide print edition of BILD did not report on this survey. Official statistics present a different perspective and receive less media attention. For example, [2] states in its Gender Equality Report that a quarter of all adult women have experienced physical and/or sexualized violence by an intimate partner at least once. According to official police statistics, one-sixth of all registered victims were affected by violence within partnerships [3].

Three aspects deserve further attention in this context. First, do the figures reported in the three sources refer to the same concepts and are they comparable? Second, are there other reasons that contribute to substantial differences in the reported numbers? Third, how were these results presented to the readers and how much attention did they receive?

Let us start with the first aspect. Although the figures appear related at first glance, a closer look reveals differences in the populations considered (men, women, couples) and in the underlying concept of violence, which may explain the striking discrepancies. The aspect of conceptualizing data is not the focus of our analysis, but also deserves attention in reporting results based on surveys.

Regarding the second issue, in the fine print, Plan International indicates that the published statement is based on an online survey conducted by a market research institute, comprising 1,000 men and women surveyed on the topic of masculinity. The study claims to be representative in terms of gender, educational status, location, and age within the 18- to 35-year-old cohort. By focusing on a particular age group likely more sensitive to the topic, the survey may have amplified the perceived severity of the findings, making them more appealing for headlines. However, it is important to note that this approach could have introduced a strong self-selection bias among participants, particularly within an online survey format. This bias occurs when individuals within the targeted age group are more inclined to participate when they see a higher relevance of the issue for themselves, possibly skewing the results and limiting their generalizability to the broader population. In addition to self-selection bias, certain groups might not have had the opportunity to participate in the survey, such as young men not included in the pool of respondents of the respective market research institute. This results from the use of a non-random sample in the study, while random sampling ensures that every element of the target population has a non-zero probability of participating. Consequently, the representativeness of the sample is compromised, affecting the validity of the conclusions with respect to the population of interest drawn from the data.

Regarding the third question, there appears to be little empirical evidence on which survey-based results make it into the news, beyond isolated case descriptions. Furthermore, for published results, additional information about the statistical framework of the analysis, e.g., the questionnaire or sampling information, is not always provided. Again, not much is

known about when additional information is provided and whether this might be taken as positive evidence for a higher quality of the performed analysis. Finally, there is no evidence that either the perceived quality (by the media) or the actual scientific quality of the survey findings influences their likelihood of being reported in the media.

To close these knowledge gaps, we address the following two goals. First, we describe the economic incentive structure for the media to use survey results and refer to them when taking survey quality and survey costs into account.

Second, we also want to take into account the level of "statistical literacy" introduced by [4]. We focus on the "knowledge component" of statistical literacy, which is relevant not only for understanding and use of statistical information in all-day situations, but also for the appropriate use of survey data and their results within the context of news production by the media [5]. This includes skills to read and understand statistical data and their visualization, as well as statistical knowledge on sampling, the research process, and statistical inference. We refer to this conception by the term "statistical literacy" in the remainder of the paper and discuss it in more detail in the section "Impact of statistical literacy". The components of statistical literacy introduced allow us to anticipate survey errors [6,7] and therefore to avoid misinterpretation of survey results when used for media reports. Other authors proposed the terms data literacy [8], as well as quantitative and risk literacy [9], which are broader, related concepts. The level of statistical literacy possessed by different actors in the media and by the public as its audience differs, which may influence the quality of the studies selected for reporting. Addressing these issues in a common framework might serve as a basis for a subsequent empirical analysis.

Our analysis focuses on the use of survey statistics in the mass media, because, particularly in Germany (our application case), the mass media are recognized as widely circulated news outlets with large audiences and significant agenda-setting power, relevant for the formation of opinions of citizens [10]. In addition, different mass media, such as newspapers and television networks, share a certain stock of news, which are also recognized in alternative media.

The paper is structured as follows. In section "Economic incentives", we propose a theoretical framework of the economic incentives faced by the actors in the media production to report high-quality survey results. The section "Impact of statistical literacy" discusses the same process with respect to statistical literacy of actors in the media and public, i.e., their abilities to assess both the quality of methods and potential survey errors. In section "Insights into empirics", we deal with empirical literature on reporting about survey results in the media and provide results of a small-scale empirical pilot study. Section Discussion summarizes the findings, points out limitations, and indicates how future empirical research based on more comprehensive data and natural language modeling could quantify the impact of different actors in our framework.

## Economic incentives

In order to investigate the pathways through which the results of surveys make their way to media reports and headlines, we will analyze in turn the roles and incentives of various actors involved, as illustrated in Fig 1. This illustration and the subsequent discussion should be considered as a general theoretical framework that illustrates the relationships between surveys as producers of statistics and their use in the (mass) media. The aim is not to cover every relevant role within the media landscape with his/her relevant backgrounds, but to introduce a structure to explain the economic incentive mechanisms when using references to survey results.

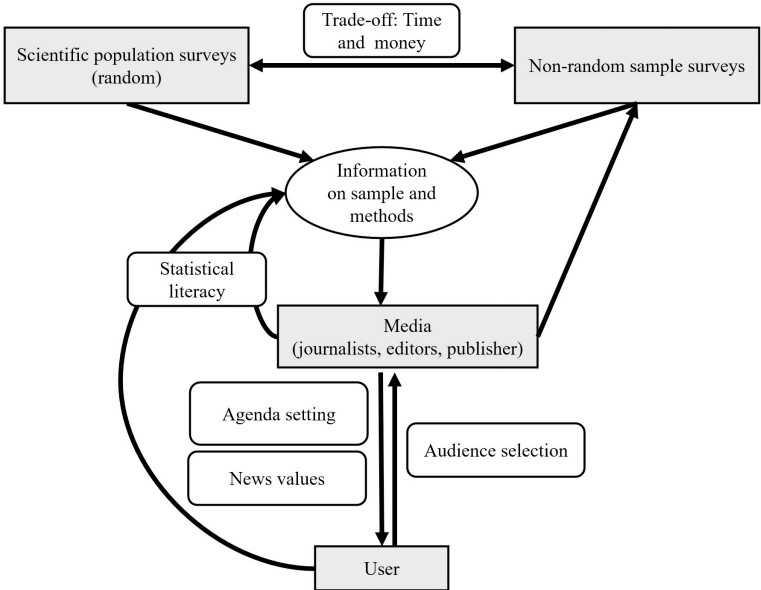

**Fig 1. Sketch of a theoretical framework on the question "Which survey results are published when, how, where and why?"** The gray boxes represent groups of heterogeneous actors involved in producing the surveys, disseminating the results (Media) and receiving them (User) (own representation, for details and references see text).

**Surveys.** Let us begin with the surveys themselves. Groves et al. [7,p. 2] define a survey as "a systematic method for gathering information from (a sample of) entities for the purposes of constructing quantitative descriptors of the attributes of the larger population of which the entities are members". Surveys are conducted through various methods such as interviews, phone calls, mail, and online platforms, with the objective of obtaining descriptive or analytic statistics and extrapolating them to the population. Surveys come in different types; the most extensive form is the census, also known as population count. Methodologically, a census involves a complete enumeration of the characteristics of the entire population. However, surveys often rely on samples, which are (small) subsets of the population, as they are more practical and consume less time and resources. Nonetheless, the ultimate goal of sample-based surveys remains to make inferences about the entire population. To achieve this, it is crucial that the survey data can be generalized to the population of interest, which is referred to as 'representativeness' [7]. Probability samples account for this aim, as every member of the population has a non-zero and known probability of being included as a feature of the study design. Reference from survey data to the population of interest therefore requires drawing a random sample from the population and understanding the process of sample selection, while also acknowledging and considering potential biases.

Consequently, we distinguish between scientific population surveys that meet these requirements and those based on non-random samples [11] that are generally not suitable for scientific use to describe populations [7]. Examples of data sets based on random sampling include the European Labor Force Survey [12], the Socio-Economic Panel [13] in Germany, and the National Health Interview Survey [14] in the US. Collecting this type of survey data is expensive and takes time. Thus, substantial funding is required, and results typically become available only several months after the survey process has started. Consequently, whenever fast responses are required or only limited resources are available,

surveys based on non-probabilistic quota or occasional samples are preferred, as they are often conducted by market research firms or other commercial entities [11] such as IQVIA (www.iqvia.com), Gartner Inc. (www.gartner.com/en), and GfK (www.gfk.com) to name a few key players in the field.

In addition to choosing the type of sample, several other factors affect the quality of data resulting from surveys, such as questionnaire development, pre-testing, survey mode, and interviewer supervision [7]. Like the choice of survey samples, decisions about these factors are made at the discretion of different actors within the institution conducting the survey or its subcontractors. The actors involved have different incentives, both directly economic and in terms of their reputation. The existing information asymmetry between these actors and their customers makes monitoring more difficult [15], which could negatively affect the quality of the results.

However, understanding these aspects is essential to evaluate potential survey errors [7] and to appropriately limit the use of the results. Survey errors particularly affect data quality in the fundamental dimensions of objectivity, validity, reliability, and representativeness. Therefore, for media actors to adequately assess survey results, they must be willing and able to obtain methodological details and communicate key quality indicators to readers. However, these goals are in conflict with the cost associated with conducting a survey or obtaining detailed information on a survey [7].

For instance, a non-random sampling framework typically comes at lower cost than a random sample. If this information is not available or ignored, the results obtained for the sample can be erroneously generalized to the entire population. The study on male violence referred to in the Introduction provides a typical example. Furthermore, it is possible for implications to be mistakenly assessed as causal, even when such causality is not supported by research methods, and the result merely indicates a correlation. Therefore, we can differentiate between scientific population surveys that use random samples and data collection methods with rigor to minimize survey errors on the one hand and non-random sample surveys on the other hand that do not rely on well-founded survey methodology to control and document survey error, for example, when relying on convenience or self-selected quota samples (Fig 1).

**Media.**   When moving down in Fig 1, following the institutions and actors involved in conducting and analyzing the surveys, the box "Media" emerges. It refers to all institutions and agents that constitute the mass media, which are communication tools that convey content to a broad audience through technical reproduction and distribution using text, images, or sound. Mass media encompasses print media (e.g., newspapers, books, brochures, posters), audiovisual media such as film, radio, and television, widely distributed storage media, and websites on the Internet [16]. Thereby, we do not claim to provide a full picture of all roles and actors within these institutions, but rather a first idea of the influence economic incentives and the statistical literacy might have.

Economic incentives for media agents vary by role, but ultimately stem from the overarching objectives of their respective institutions. These incentives differ depending on the revenue model, be it subscription-based, reliant on individual purchase decisions, or funded through paid advertising.

In general, economic performance will depend on reaching as many people as possible. Our focus is on the media that employ text, specifically the print media and their websites. In contrast to print media, which seeks to maximize the number of copies sold, websites on the Internet strive to maximize the number of clicks on their articles [17].

Beyond economic incentives, the media, and particularly the mass media, are expected to collect, frame, and disseminate relevant information in different areas of political, economic, and societal life to the public. Thus, the media play a crucial role in influencing and shaping

public opinion [18–20]. Consequently, the media play an important role in the dissemination of survey results to the public. News media aim to provide trustworthy information that enables audiences to form informed opinions and decisions. In doing so, media outlets claim to be objective, neutral, and critical [21]. However, it should always be kept in mind that the media including a large number of different actors with different background and individual aims are subject to economic constraints and incentives.

In theory, Lippmann [22] initially introduced the concept of news value to explain why journalists choose to cover certain events while overlooking many others. He posited that events become newsworthy due to specific attributes, later termed news factors, that make them stand out. Over time, the theory of news selection has made significant advances. Östgaard [23] delved into the discussion of three categories of news factors – simplification, identification, and sensationalism – that contribute to the newsworthiness of an event. Meanwhile, Galtung [24] went one step further by defining 12 news factors. Schulz [25] took a different approach by defining news values as hypotheses journalists form about the relevance of events. He developed scales to gauge the intensity of news factors and used the placement and length of news stories as indicators of their news value.

More recently, the two-component theory [26] has offered an explanation and prediction of the newsworthiness of stories, which is the likelihood that these stories are chosen for publication. Central to their theory are news factors and news values. News factors represent the characteristics of news stories and can vary in intensity. Their collective impact on the selection of news stories is referred to as the news value. Unlike news factors, news values are not inherent qualities of news stories, but reflect the judgments made by journalists regarding the relevance of these factors. However, in the two-component theory, news factors and values are not the sole determinants of newsroom decisions. Other factors, such as the editorial stance of a newspaper, general preferences for specific topics, individual preferences of journalists, and external pressures from politics and business, are relevant, among others. According to the Agenda-setting theory [19,20,27], topics and information salient in the media are likely to be important for the public and may shape public opinion. Priming theory [28] indicates that the attention of the media to an issue causes people to put a special weight on this issue. Therefore, studying which topics are reported in the media and with which precision is crucial in understanding public opinion and its dynamics.

In reality, the selection criteria in (quality) journalism and in research are fairly similar, as they seek to identify topics that are true, new, relevant and likely to resonate with their audience [29]. The more an event satisfies these conditions, the more likely it is to be selected as news [30]. High-quality survey results are often lacking for time-sensitive topics, due to the lengthy process required to conduct comprehensive and probability sample studies. Such studies require a considerable investment of time and resources, leading to a time delay in the release of pertinent data. In addition, the data analysis process further exacerbates this temporal discrepancy. Consequently, media organizations employ pragmatic strategies, such as enlisting commercial service providers to execute surveys on their behalf, or, in some instances, conducting surveys autonomously, as a means to address this challenge. These investments sometimes include long-term high-quality probability sample surveys. For example, the institute "Forschungsgruppe Wahlen" conducts such phone surveys based on a probability sample for the "political barometer" of the German TV broadcaster ZDF. To obtain timely results, especially during election campaigns, media outlets frequently rely on less comprehensive polls based on online samples. These surveys often lack accompanying information on data quality.

**User.** The last agent considered in Fig 1 is the user of the media. In the past, the prevailing notion was that the user existed somewhat outside the sphere of influence when it came

to determining the topics covered in the daily newspapers. The editorial decisions seemed insulated from direct user input or feedback, positioning the audience as passive recipients rather than active participants in shaping the news agenda. The traditional belief was that journalists and editors dictated content without immediate regard for the specific preferences or interests of the audience. However, this landscape has changed. User-centric approaches now acknowledge and incorporate audience preferences in shaping both print and digital content. Wendelin et al. [31] explore current differences between journalistic news selection and selections of the audience using user rankings. The results show similarities in news values, but differences in preferred topics. Although this user-centric approach empowers users with more control over their news consumption, it also challenges news organizations to balance audience demands with journalistic integrity and societal impact.

## Impact of statistical literacy

So far, we have delineated some actor groups involved in the process of survey results transitioning into news within the media. Reversing the perspective in Fig 1 raises the question of whether journalists and readers alike are able to interpret and contextualize survey results accurately, including their nuances and implications, which will be addressed in this section.

Before addressing the role of statistical literacy in the context of media reporting and public understanding of survey-based results on socioeconomic topics, we first have to introduce the concept of statistical literacy itself, which might be seen as a subset or derivative of the broader idea of scientific literacy. These two concepts will be introduced in subsections Scientific Literacy and Statistical Literacy.

Subsequently, the third subsection Interaction of Incentives and Statistical Literacy will apply these concepts to the framework introduced in the section "Economic incentives".

### Scientific literacy

The OECD [32] defines scientific literacy as "the ability to engage with science-related issues and with the ideas of science, as a reflective citizen". Scientific literacy integrates aspects of literacy in writing, numeracy, and digital skills, focusing on the comprehension of science, including its methods, measurement techniques, units, observations, and theoretical frameworks. It also involves foundational knowledge in key scientific domains like natural sciences and computer science. In general, the concept of scientific literacy refers to public understanding of science, which is the result of the knowledge, beliefs, and opinions of citizens about science [33]. Scientific literacy reflects the level of scientific education and competence in collecting and analyzing information. It is a prerequisite for informed citizens' decisions, e.g. voting for a political party or accepting governmental measures [33].

Referring to Fig 1, scientific literacy is pivotal at two junctures: among the media and journalists, as well as the end users. Stephen et al. [34] state that "Since most people do not read raw data-sets, journalism has a key role to play in the discovery, translation, and interpretation of statistics". This highlights its significance in both the dissemination and the understanding of scientific information. The media often report on societal and economic issues and aim to present a reliable picture of the current situation, frequently using scientific information to support their claims [21]. Thus, journalists' ability to select, prepare and communicate scientific results – as well as their methodological knowledge –is crucial to evaluate the quality of referenced data and studies. Meanwhile, the informed public should be provided with information that allows an assessment of the quality of scientific and statistical data.

In addition to journalists, scientific literacy of the public is also of interest in the present analysis. In past research, it was found to exhibit some deficiencies, but more recent evidence on this aspect appears to be scarce. For example, an early US study found that only 10% of the participants agreed with the statement that doing science involves implementing systematic and controlled variation [35]. In a study among more than 2,000 British adults in 1988, only 34% of the participants correctly responded that the Earth orbits the Sun once a year and 28% knew that antibiotics kill bacteria but not viruses [36]. Americans' confidence in scientists has been found to be high for approximately 40 years [37]. Scientific literacy correlates with political knowledge, as well as with trust in science and institutional trust [38]. Little trust in science was found to be associated with the neglect and rejection of established scientific knowledge and the acceptance of conspiracy theories [39].

## Statistical literacy

Statistical literacy is a subdomain of scientific literacy [33]. It refers specifically to the statistical knowledge and reasoning required to interpret statistical information and solve related problems in everyday situations. Gal [4] describes statistical literacy – in a similar vein to scientific literacy – as one of the basic competencies that are relevant to deal with information flow and to participate in societal life. Like scientific literacy, statistical literacy is conceptualized as comprising two so-called components: "knowledge" and "dispositional" elements, both relevant for informed problem-solving, decision-making, and statistically literate action [4]. Let us have a detailed look at those two categories: Knowledge elements encompass literacy skills, including the ability to read and understand numeric data and their visual representations, as well as contextual knowledge in the form of basic scientific knowledge in the natural sciences. In addition, mathematical and statistical knowledge is crucial, covering topics such as sampling, the research process and design, and statistical inference. This knowledge category also emphasizes the importance of critically questioning the given statistical information. In contrast, dispositional elements are less clearly delineated. They include beliefs and attitudes such as the willingness to invest mental effort and the openness to engage with different methodological approaches.

When it comes to understanding surveys, the knowledge component of statistical literacy is particularly relevant. It encompasses an understanding of terms such as "data" and "variables," familiarity with the research process – including sampling, research design, and instrumentation – as well as knowledge of statistical inference and the ability to critically evaluate statistical information [40].

Statistical literacy may not only be essential for journalists when selecting high-quality survey results to report on, especially given the public's potentially limited ability to process methodological and technical information. A statistically literate journalist might also challenge a press release with dubious claims, rephrase a misleading headline, or add context that helps readers grasp uncertainty and nuance. Conversely, if we assume a higher level of statistical literacy among the public, it becomes important to contextualize the findings and critically examine how they were produced, which requires some key information on data quality to be provided.

## Interaction of incentives and statistical literacy

Scientific and statistical literacy are relevant concepts for everyone in dealing with information flows in our daily lives. Although statistical literacy levels remain low in the general population [35,36], they have shown a steady upward trend since 1989 in regions such as

Europe, Japan, China, and the United States [41]. Most users cannot interpret the survey results on their own and rely on journalists to process and communicate the information.

Upon receiving the research findings, journalists and media outlets critically assess the importance and newsworthiness of the survey results. Mass media outlets present and often translate or interpret professional research findings for the general public. The reasons why and the extent to which this occurs vary according to the type of media, ranging from the information mandate of public broadcasters, who are obligated to inform about all matters of importance, to more self-serving motives and selective reporting aimed at generating high click rates and circulation. It is important to note that during the process of turning survey results into headlines, some degree of simplification or sensationalism can hardly be avoided. This can lead to potential distortion or oversimplification of complex survey findings. Nevertheless, reputation is also a relevant asset of the media. Therefore, reputable media outlets strive to maintain accuracy and transparency in their reporting, citing original sources and providing sufficient context for the readers to grasp the nuances of the survey results.

## Insights into empirics

The considerations thus far have been theoretical, prompting us to turn our attention to empirical findings. The subsection State of the Art of Reporting Polls in the Media reviews the existing empirical literature on how polls are covered by the media. Building on existing research, we conducted an illustrative study ourselves. To this end, we considered selected press articles from DER SPIEGEL and BILD referring to surveys. The results of this small-scale empirical pilot study are presented in the second subsection Study on Reporting Surveys in DER SPIEGEL and BILD .

### State of the art of reporting polls in the media

Existing literature has predominantly focused on the media's portrayal of polls, particularly in the context of pre-election periods. For example, a study on how Singapore's two leading newspapers reported public opinion polls found that these surveys suffered from theoretical and methodological shortcomings, which were rarely addressed in the media coverage [42]. In Canada, a study of 1997 federal election coverage revealed a lack of basic technical information in poll reporting on television and in newspapers, such as sample size, which was rarely mentioned [43]. Similarly, the elements most commonly reported in Canada were the organizations that sponsored and conducted the survey, together with the time period in which the survey was conducted [44]. This trend of minimal disclosure, which hinders the public's ability to assess the reliability and validity of a poll, was corroborated in the United States, where the sponsor and sample size were the details most frequently cited by both national and local newspapers [45].

Reports of federal election coverage from 1980 to 2002 in four major German newspapers frequently mentioned polling organizations, commissioning bodies, and fieldwork periods [46]. In a subsequent analysis involving the same newspapers, as well as tabloid and weekly magazines, the name of the polling institute was the most frequently provided detail, although absent in 30% of the articles [47]. In Brazil, the trend of publishing an increasing number of articles based on the results of polls was noted, including some methodological information, although briefly [48]. This scarcity of detailed information was similarly reported in South African media [49] and Taiwanese newspapers and television [50]. Mateos and Penadès [51] scrutinized the coverage of electoral polls, distinguishing between those sponsored by media entities (but conducted by private organizations) and those originating from public

institutions, finding that although key methodological details were often included, they were never comprehensive.

Recent studies continue to highlight concerns; for example, in Spain, where newspapers are legally required to disclose the methodologies behind election polls, a study of articles leading up to the 2012 Catalan parliamentary elections found that only two-thirds included this information, which was at times incomplete or inaccurate [52]. Louwerse and van Dijk [53] noted a similar paucity in the quality of Dutch poll coverage, with few reports offering details on the polled universe, sampling methods, non-response rates, sample sizes, weights, or fieldwork dates. However, the data collection method and the margin of error were mentioned more frequently, particularly with regard to the 2017 elections.

## Study on reporting surveys in DER SPIEGEL and BILD

Building upon the existing body of literature on polls, we present a small-scale empirical pilot study for illustrative purposes that is specifically focused on the reporting of surveys. The intention of the study is to evaluate the information on the statistical findings provided by selected newspapers in Germany. In particular, we contrast DER SPIEGEL that commits to the accuracy of provided information by self-definition and BILD, known for repeated violations of the press code. Both DER SPIEGEL and BILD are recognized as relevant representatives of the mass media in Germany, available also in digital formats.

**Methods.**  We analyzed articles published by DER SPIEGEL online and BILD in the second half of 2023. We utilized the digital format of DER SPIEGEL and BILD, accessed the online articles through the Nexis Uni database and selected all articles published between July $1^{st}$ and December $31^{st}$, 2023.

DER SPIEGEL is a German weekly news magazine established in 1947 and accessible online since 1994. With a weekly circulation of about 700,000 copies in 2023 [54], it was one of the most widely distributed publications of this type in Europe. Renowned for its investigative journalism, DER SPIEGEL played a pivotal role in revealing major political misconduct and scandals. According to The Economist [55], it was one of the most influential magazines in continental Europe. Known for its incisive analysis and accessible approach, DER SPIEGEL often incorporated elements of popular science making it particularly appealing for our analysis.

We excluded articles that summarized the most important news of the day ("Lage am Morgen/Abend"). These overview articles occasionally provided information on surveys; however, this information is often sourced from other articles included in our sample, resulting in duplication if not excluded. From the remaining 10,185 articles we kept those including the word "survey" at least once. The keyword "survey" as well as other keywords used to characterize the content of survey-related articles were used in German due to the primary publication language of both newspapers and then translated into English for the documentation of the results. This pre-processing step resulted in 265 unique articles from DER SPIEGEL, that is, 2.60% of all published articles.

BILD has been published daily since 1952 with a circulation of still more than 1,000,000 copies in 2023. BILD is known for its sensational headlines that lead to a high number of violations of the press code.

We applied the same method for BILD as we did for DER SPIEGEL by starting with all 5,855 articles published from July to December 2023 in the nationwide printed edition. Next, we limited the sample to those 144 articles including the word "survey". Again, all keywords were used in German and then translated into English for the documentation of results. The share of articles referencing surveys in BILD was 2.46%, nearly identical to DER SPIEGEL's

2.52%. One first difference from DER SPIEGEL was given by the existence of some articles in BILD that contained the keyword " survey", but did not refer to any specific survey (9 of 144, 6.25%). For the following analysis, we excluded these articles and focus on the 135 remaining ones.

To analyze the reporting, we manually examined each article for source references, information on sample size, and the presence of statistical terms such as methodology, (non-)representativeness, statistical error, and bias. The intercoder reliability for e.g. detecting the sample size among three coders, measured as Fleiss' Kappa, was 0.75.

**Results.** DER SPIEGEL categorized its articles by themes. The content of the survey articles predominantly addressed economics (36.98%) and politics (18.11%). Furthermore, 13.96% of the articles provided information on foreign news, while only a small fraction, specifically 6 of 265 articles (2.26%), were classified under science by DER SPIEGEL itself. However, our primary interest lay not in the content, but in the surveys and their reporting.

Of 265 articles, 227 (85.66%) contained information about the source of the survey. Although most articles mentioned a source, this was often limited to the name of an institution or author, without reference to a research paper or quality report. The source most frequently cited in the research corpus was Civey, a pollster firm that appeared in 28 instances, accounting for 10.57% of all articles analyzed. This institute conducted exclusively online surveys on Internet users, focusing on Sunday opinion polls, measuring the prevailing political sentiments by asking the question "Who would you vote for if the general election were held this Sunday?". Furthermore, the ifo institute was another commonly referenced source, cited in 22 articles, representing 8.30% of the total. Most of these articles discussed the ifo Business Climate Index, a monthly economic indicator that evaluated the economic growth prospects in Germany. Slightly more frequent were the surveys conducted by DER SPIEGEL itself with 23 articles (8.68%). These articles often included infographics that offer additional information, such as sample size.

When referring to surveys, 116 articles (43.77%) provided further information on the number of participants, ranging from 9 to 100,000 with a median value of 2,011. With regard to the statistical methodologies used, the provided data was notably sparse. 58 articles (21.89%) provided insight into the statistical background, e.g. representativeness, error rates, or other methodological issues. Although 29 articles, comprising 10.94% of the total, declared that the underlying research was representative of a specific part of the population, none of them delineates the background sampling procedure. Furthermore, in 19 articles (7.17%), additional methodological details were accessible through a link to the source's website, exclusively the firm Civey in these cases. In other seven articles (2.64%) statistical considerations such as potential biases, averages, or errors were addressed. In some cases, additional information was provided only as part of an infographic.

To verify the primary sources referenced, we randomly selected seven articles. The corresponding research papers were easily located in four instances. For another article, while the specific research paper was not recovered, the underlying survey and its findings were successfully identified. However, in the case of the remaining two articles, the definitive identification of the sources proved elusive, although supplementary information was obtained directly from the authors. This highlighted the varying degrees of source transparency and accessibility in journalistic reporting.

With these insights, all articles in the sample were classified into four groups based on the level of information they provided about the surveys referenced.

- `full information`: includes the source, number of participants, and statistical keywords (representative, error rate, sampling procedure, ... ).

- `source +1`: entails either the number of participants or statistical keywords (representative, error rate, sampling procedure, ... ), alongside the source.
- `only source`: it refers solely to the source of the information.
- `no information`: lacks any mention of the source or additional survey details.

Fig 2 shows the distribution of articles in DER SPIEGEL among the four groups by the blue bars. Of the total corpus of 265 articles, 38 articles (14.34%) contain no information whatsoever on the basis of the statements they present. In contrast, 227 articles (85.66%) at least cite a source. Within this subset, 122 articles (46.04%) provide the source along with at least one additional piece of information, while only 48 articles (18.11%) offer comprehensive details, including both the number of participants and insights into the statistical methods or associated keywords used.

To further illustrate the variation in the reporting details, we consider the first articles (by publication date) in our sample with `no information` and `full information` according to our simplified classification. The first article without any further information on the survey was published on 8 July 2023. It deals with leaders of federal states from the social-democratic party who criticize the climate policy of the federal government. In one sentence in the article it says "In Thuringia, a recent survey even places the AfD at 34%". In the article, which comprises 374 words, no further information on the survey is included. The second article, published on 8 July 2023 entitled "The large majority think that capping parental allowance is right" (510 words) states that the survey was conducted by Civey on behalf of DER SPIEGEL. An infographic in the article provides information about the field period, the sample size, and the statistical imprecision of the reported shares. Furthermore, a link is provided for a further detailed description of the methodology used by Civey that does not use probability sampling, while including a statement about representativity. However, the content of this link was changed in March 2024, when DER SPIEGEL ended cooperation with Civey.

The orange bars in Fig 2 summarize the findings for BILD. 117 of the 135 articles (86.67%) included at least information about the source of the survey. With 34 explicit mentions, INSA

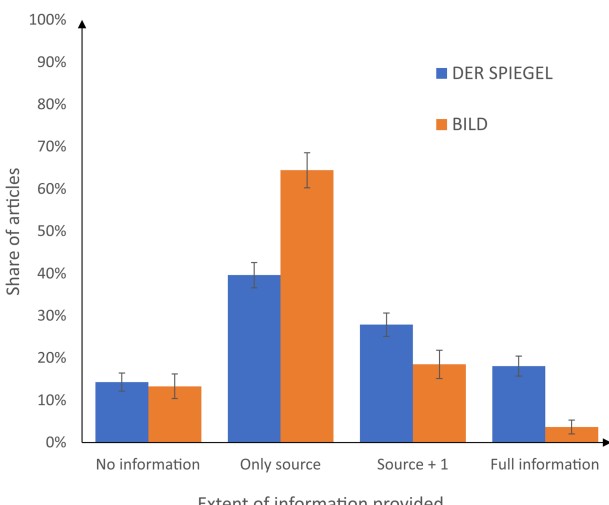

**Fig 2. Categorization of articles based on the extent of survey information disclosure for DER SPIEGEL (blue) and BILD (orange) (percentage among all articles that refer to a survey).** Error bars show ±1 sample standard deviation of shares.

Consulere covers about a quarter of all sources (25.19%). For 27 of them, BILD was mentioned as the initiator of the survey. For another eight surveys, BILD was named as executor of the survey. Only in 30 articles (22.22%), additional information on the number of participants was provided (20 to 235,000 with a median of 1,072). 14 surveys (9.82%) were labeled as "representative" in the article, but further explicit information on the sample and methodology was provided only in 5 articles (3.7%). In summary, detailed information on surveys seems to be more scarce in BILD, which might have been expected given that the length of the article is also typically much shorter compared to DER SPIEGEL. However, other factors discussed in our framework may also contribute to these differences.

As for DER SPIEGEL, we briefly report on the first two articles in BILD with `no information` and `full information`, respectively. The first article entitled "Winner" and published on 8 July 2023 states that "According to a new major poll, the 20-minute performance by Freddie Mercury [...] was voted the most memorable festival moment of all time". No further information on the survey was given in the 52 words of the short article. The second article published on 6 July 2023 entitled "How AfD is Germany?" (1035 words) refers to two surveys contacted by INSA Consolere on behalf of BILD. It also provides information on the field period, sample sizes, and a statement about representativeness that referred to a quota sample.

## Discussion

In this paper, we present a theoretical framework for reporting survey results in the media. In this context, we discuss the role of the main actors involved at a relatively broad level of abstraction.

We propose that the use of scientific and survey data or informed and critical reference to them requires a sufficient level of statistical literacy, which may exceed the average level found in the general population. Furthermore, economic (dis)incentives affecting media actors could limit appropriate reporting. Furthermore, we conducted a small-scale empirical pilot study to illustrate how survey results were reported and which methodological background information was provided. We used two relevant German outlets DER SPIEGEL and BILD, which reach a large audience, but differ in the compliance to the journalistic principle of the accuracy and scientific foundation of the presented news. We focus on the media due to their relevance in educating citizens by providing high-quality information and its role in shaping public opinion [19,20].

In both outlets, we found a small and comparable share of articles that referred to survey statistics. The results indicate a missing or imprecise use of statistical references or the provision of statistical information without context. In particular, the information on representativeness was less sufficient, as the surveys cited were online surveys without the use of probabilistic procedures for sample selection. Both outlets, DER SPIEGEL and BILD, provided mainly information on the source of the survey, but hardly additional information on the methodology relevant to understand the credibility of the surveys.

In general, the information provided by journalists or the media about the surveys was quite limited. Journalists and individuals responsible for news production should therefore be trained in science methodology and statistics, at least to a level allowing them to evaluate the credibility of data and studies they select for their references. Furthermore, media producers should be able to communicate not only the content of the studies but also relevant information related to their quality.

Therefore, the results highlight the need for better training of responsible actors in statistical literacy. In addition, the observed issues of selective and sloppy use of statistical

references, or providing statistical information without context, can partly be attributed to the pressure of reporting interesting news (sensation seeking) as well as time and budget constraints.

In the following, we discuss some limitations of the empirical study provided and bring them into relation to the proposed theoretical framework, particularly to be able to identify relevant questions for further research.

First, based on our limited evidence, we are not yet able to draw conclusions about whether limitations in statistical literacy in the broad sense and/or economic (dis)incentives are the main drivers of our results. Thus, further research would have to address the incentive structure for all actors working in this domain in more detail. Only a combination of appropriate incentives and adequate levels of statistical literacy of media professionals will help to improve the quality of survey reporting in the media and thus ensure that the public receives accurate and contextually sound information. Although a more detailed analysis of the incentive structure seems feasible at the theoretical level, finding appropriate data for hypothesis testing appears to be a challenge.

Second, our analysis suggests that while we can theoretically describe the effects of selection and relevance mechanisms in the reporting of surveys by the media, we do not yet fully understand the magnitude and impact of these effects and how they are triggered by different actors involved in the process. Therefore, there is significant potential for further theoretical development and empirical research in this area. Future studies should aim to quantify these selection effects and investigate the factors that influence the decision-making process of different actors in the media, also depending on the type of media outlet when choosing to report on certain surveys or to commission surveys themselves. This will help to understand the extent of biases and limitations inherent in the media reporting of surveys.

Third, the empirical analysis should cover more relevant mass and possibly also social media as well as longer time periods. This will allow for both the identification of factors at the media level and time trends in the selection of and reporting on surveys. In this context, to allow for the analysis of larger corpora of news data both with regard to longer time periods and a broader selection of news outlets of different types, it is essential to develop automated methods for coding and analyzing media content. Currently, as in our small-scale empirical pilot study, the process of screening articles for specific information often relies on human coders, which is time consuming, expensive, and subject to human error or bias. For clearly defined characteristics, such as sample size ($N$) or the source of the survey, keyword searches could simplify the coding process. However, for more complex statistical measures that require contextual understanding, more sophisticated automated tools need to be developed. Hence, exploiting recent advances in natural language processing (NLP) appears to be a promising avenue for the analysis of larger news corpora. We explore such methods, e.g. using large language models (LLM) to extract relevant information in a semi-automatic way. The initial findings are promising. Thus, this will be the next step in our future research.

Moreover, further scientific research on statistical literacy of populations, among others in Germany and Europe, is required to extend the analysis presented in this article to the perspective of recipients of media news. This can enrich studies on statistical literacy of mass media makers and journalists and help to critically reflect current practices.

## Acknowledgments

The authors thank Theresa Daniel and Maximilian Herr for their valuable research support and two anonymous referees for valuable comments on previous versions of this paper, which helped to improve its presentation.

## Author contributions

**Conceptualization:** Jenny Bethäuser, Natalja Menold.

**Data curation:** Jenny Bethäuser.

**Formal analysis:** Jenny Bethäuser.

**Methodology:** Jenny Bethäuser.

**Supervision:** Peter Winker.

**Validation:** Jenny Bethäuser, Natalja Menold.

**Visualization:** Jenny Bethäuser.

**Writing – original draft:** Jenny Bethäuser, Peter Winker, Natalja Menold.

**Writing – review & editing:** Jenny Bethäuser, Peter Winker, Natalja Menold.

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
