## [Decision Letter · Decision Letter 0]

31 Mar 2025

PONE-D-25-03801The Impact of Statistical Literacy and Economic Incentives on the (Mis-)Use of Survey Based Statistics in Media Reporting - A FrameworkPLOS ONE

Dear Dr. Bethäuser,

Thank you for submitting your manuscript to PLOS ONE. After careful consideration, we feel that it has merit but does not fully meet PLOS ONE’s publication criteria as it currently stands. Therefore, we invite you to submit a revised version of the manuscript that addresses the points raised during the review process.

We look forward to receiving your revised manuscript.

Kind regards,

Felix G. Rebitschek

Academic Editor

PLOS ONE

Journal Requirements:

Additional Editor Comments:

Dear authors,

Thank you for your important contribution to the presentation of survey-based statistics in the media! You have received two largely constructive reports. By addressing the concerns expressed in them, the manuscript should benefit significantly.

Sincerely,

FR

Reviewers' comments:

Reviewer's Responses to Questions

**Comments to the Author**

1. Is the manuscript technically sound, and do the data support the conclusions?

Reviewer #1: Partly

Reviewer #2: Yes

2. Has the statistical analysis been performed appropriately and rigorously? 

Reviewer #1: Yes

Reviewer #2: I Don't Know

3. Have the authors made all data underlying the findings in their manuscript fully available?

Reviewer #1: Yes

Reviewer #2: Yes

4. Is the manuscript presented in an intelligible fashion and written in standard English?

Reviewer #1: Yes

Reviewer #2: Yes

5. Review Comments to the Author

Reviewer #1: The topic and the research question addressed in the manuscript is of enormous relevance in a world in which more and more statistics are disseminated through the media and in which there is a great demand for statistics that are both good and well reported. So far, there is a lack of models that adequately describe the statistical literacy of various actors in creating media products. Therefore, I welcome the presented work. However, there are two major aspects I want to address, and which will lead to my recommendation for (very) major revision: 1) There is a need for a more detailed view on “the media” regarding to, which group of people needs “statistical literacy” and what kind of “statistical literacy”; this might differ regarding to the type of media and different group of people (see below); 2) The focus on “SPIEGEL” is too narrow and should be extended (see below). These aspects (and also further minor comments) require a very major revision of the article, which I believe would be possible. I therefore rate the article as requiring a very major revision, but I would be very happy to read a revised version of the article.

Two disclaimers: I review the manuscript from a perspective of statistics education, and I am not an English native speaker!

Major issues:

1) You are speaking about “The media” in a very general way. However, different group of people are working on medial products. For example, journalists with MANY different ways of becoming a “journalist”. There are (sometimes) also data journalists and science journalists, people who finally check the medial products, persons/institutions who conduct the study and so on. Given the aim of the article, one would hope for a much more detailed and differentiated consideration of all these players. After all, it is precisely these players who need “statistical literacy”. Depending on their role and the media product (SPIEGEL vs. BILD vs. Spektrum der Wissenschaft vs. NY Times), their knowledge requirements may vary greatly!

2) Why do you focus only on “SPIEGEL”. What would change, if we look at “BILD”, “Spektrum der Wissenschaft” or international news papers or media? In my opinion, this focus is much too narrow to achieve a good picture. I would find the results much more exciting and valid if they were to expand them to include other (even very different) media. Because with this very narrow view, no generalization is possible and I wonder what the results mean at all.

Further issues:

“Statistical literacy”: Do you really mean statistical literacy or data literacy? These concepts would have to be described in more detail, distinguished from one another and consciously chosen. There is a wealth of literature available that explains the comparisons between these two areas.

Abstract: “Statistical literacy” of whom? Of the reader, of the person who conducts the study, presents the results, of the journalist ….? Please highlight in the abstract which groups of people are involved.

Introduction: “The first concerns the potential explanations for the discrepancies observed among the figures – specifically, the variation between one-third, one-fourth, and one-sixth” This comparison is very problematic (not only due to the two aspects you further describe in the paper. The three different proportions are related to three different questions, very sloppy formulated: “Proportion of men, who have resorted to physical violence“, “Proportion of women, who get a victim of domestic violence”, “proportion of violence within a partnership under victims of domestic violence”. Even though I understand that such a comparison of very different percentage rates would be attractive, it is clearly misleading because it does not look at the same proportion or the same question. The ability to distinguish between such things would also be immensely important for readers of media reports.

Figure 1: The arrow from media with “statistical literacy” does not fit. Here, as with user, a person or group of people should also be named. And that is precisely what would be interesting. WHO is involved in the chain here? Who all needs statistical literacy? And what kind of statistical literacy in each case?

Introduction: When defining the concept of statistical literacy, please provide a citation.

Allover the manuscript: Whenever you are speaking about “statistical literacy”, please clarify the person or group of people, who needs statistical literacy, because in the whole process of conducting the study (or the person or institute who commissions the study) to reading the news, there are a lot of people who need statistical literacy.

Page 5 – line 201: “Until now, we have delineated the individual actors” No, you have not; see comments above.

The article “Zhang H, Wang H. Information skills and literacy in investigative journalism in the social media era. Journal of Information Science. 2022; p. 447–465. doi:10.1177/01655515221094442.” you cited, has been retracted from the journal. I don’t know, what might be an appropriate way of cite this article (or if it is even possible to cite this article any more).

Smaller Issues:

- Page 2: “Section Economic Incentives , ” Please delete the space before the comma.

- Page 3: “In Section Insights into Empirics ,” the same

Reviewer #2: The Impact of Statistical Literacy and Economic Incentives on the (Mis-)Use of Survey

Based Statistics in Media Reporting - A Framework

The paper provides a conceptual framework on how statistical literacy and economic incentives influence presentation and understanding of survey data for media reports. The authors clearly lay out the methodological shortcomings, principles of survey design and its biases, and provide a conceptual framework that could perhaps serve as a foundation for future empirical studies to better understand the several dynamics at play while evaluating media reporting.

I think this is extremely critical topic given the data intensive world we are all in. Also, there is no doubt the conceptual framework provides a sense of direction for future research in this area. I have a few comments and hope these reflections shall strengthen the manuscript.

Also, the draft is well organized in terms of writing, and I enjoyed reading it!

1. I recommend the authors to clarify terms such as "statistical literacy" early in the manuscript to ensure consistency, and with appropriate definition that might be relevant for journalism. For instance, is it just the problem of quantitative literacy? That is, would numeracy literacy solve the problem? Or, it is more pervasive as it includes the relevance of risk literacy as well.

2. I encourage the manuscript to provide concrete examples of the ways in which news reports systemically misinterpret, misrepresent, or misuse numerical data as part of the reporting process, and probably ranging across diverse topics, such as healthcare, war, finance, elections, poverty, etc. The point is that, for the world of Journalism, the issue of statistical literacy is more pervasive and fundamental, than for a health-care expert or a finance expert, where it could be more topical and functional. To elaborate, the issue of reporting relative versus absolutive risks is very topical in Healthcare.

3. Though, the manuscript is purely conceptual and it does not include empirical evidence to validate its proposed framework. Incorporating preliminary data or case studies would strengthen its claims. Also, if it would be possible to analyze trends in media reporting using a dataset of survey-based articles to identify patterns in statistical misrepresentation.

4. While the paper discusses economic incentives driving media behavior, it does not delve deeply into specific mechanisms (e.g., advertising revenue models or audience engagement metrics). A more detailed analysis would enhance understanding. Also, if this relation could be elaborated through case studies or through an empirical setting.

5. With regards to the issue of economic incentives, I am also pretty curious to know if there are previous studies that clearly establish that statistical literacy would lead to more accurate reporting and a more socially-responsible journalism? I ask the authors to think about this question because a discussion on the trade-offs between the quality of survey-based statistics and the economic profitability would be interesting to reflect. Would the association btw statistical literacy and improved media reporting be so simple?! Although, the authors sparingly mention about the trade-off between time and money in the money.

6. Another point to reflect is that of the economic incentives for different actors -- I know the authors have discussed it briefly, however, a much detailed discussion on role of survey providers, advertisers, and other actors in the supply chain of information generation including the public who consume it, would be an important conversation.

I think economic incentives are an important influence, it is just not discussed well in the manuscript.

Minor comments:

7. Have the authors thought about including "Subjective Numeracy" in the conceptual model? This is particularly interesting to allay concerns about the response burden and user acceptability of the test-like mathematical problems included

in objective assessment instruments.

8. In Figure 1, lacks proper citations, especially around variables from prior literature.

6. PLOS authors have the option to publish the peer review history of their article (what does this mean?). If published, this will include your full peer review and any attached files.

Reviewer #1: No

Reviewer #2: No

---

## [Author Response · Author response to Decision Letter 1]

7 May 2025

Response to Reviewers for Article

“The Impact of Statistical Literacy and Economic Incentives on

the (Mis-)Use of Survey Based Statistics in Media Reporting

– A Framework”

May 5, 2025

We are indebted to the reviewers for their detailed comments and suggestions regarding

our paper and to the editor for inviting us to revise the paper.

We appreciate that both reviewers consider the topic highly relevant. We also agree with

them that the presented modeling framework and initial empirical insights are just a first

step, which must be followed by further empirical research. This future work will have to be

based on a broader dataset, which should include different media outlets as well as longer

time spans. However, this follow-up study will also require novel methods such as natural

language processing (NLP) to allow a proper analysis with our limited human resources.

Thus, this extension of the analysis, also suggested by the reviewers, is beyond the scope

of our revision. However, we were able to extend our empirical illustration by collecting

a dataset from the tabloid BILD for the same time period as the existing data for DER

SPIEGEL.

In the following, we will explain how we responded to the specific comments made by the

reviewers. The original comment is repeated first, followed by our response in italics.

1 Reviewer #1

The topic and the research question addressed in the manuscript is of enormous relevance

in a world in which more and more statistics are disseminated through the media and in

which there is a great demand for statistics that are both good and well reported. So far,

there is a lack of models that adequately describe the statistical literacy of various actors

in creating media products. Therefore, I welcome the presented work. However, there are

two major aspects I want to address, and which will lead to my recommendation for (very)

major revision: 1) There is a need for a more detailed view on “the media” regarding to,

which group of people needs “statistical literacy” and what kind of “statistical literacy”; this

might differ regarding to the type of media and different group of people (see below); 2) The

focus on “SPIEGEL” is too narrow and should be extended (see below). These aspects (and

also further minor comments) require a very major revision of the article, which I believe

would be possible. I therefore rate the article as requiring a very major revision, but I would

be very happy to read a revised version of the article.

We thank the reviewer for highlighting the relevance of our research question, which

obviously can only address some aspects of the overarching theme of statistical literacy in

the media. We fully agree with issue 1) and try to be more precise about the specific actors

involved and the type of “statistical literacy” required depending on their role. However, we

cannot claim to describe all details of the process, which might be a relevant subject for future

research.

We also agree with your second statement and have added additional results for a tabloid

newspaper in our analysis. Moreover, we discuss the limitations of our empirical illustrations

and suggest extensions for an extensive empirical analysis as a follow-up to the present

conceptual framework. We have already started to develop methods from natural language

processing (NLP), using, e.g., large language models (LLM) to extract relevant information

in a semi-automated way, but given the deadline for the revision, we have to leave this aspect

to future research.

Two disclaimers: I review the manuscript from a perspective of statistics education, and

I am not an English native speaker!

We are fully aware that our interdisciplinary paper might be considered from different

angles. Thus, we appreciate your perspective on statistics education very much.

Major issues:

1) You are speaking about “The media” in a very general way. However, different group

of people are working on medial products. For example, journalists with MANY different

ways of becoming a “journalist”. There are (sometimes) also data journalists and science

journalists, people who finally check the medial products, persons/institutions who conduct

the study and so on. Given the aim of the article, one would hope for a much more detailed

and differentiated consideration of all these players. After all, it is precisely these players

who need “statistical literacy”. Depending on their role and the media product (SPIEGEL

vs. BILD vs. Spektrum der Wissenschaft vs. NY Times), their knowledge requirements may

vary greatly!

As pointed out above, we fully agree with your argument. Consequently, we tried to

describe the roles in more detail and highlight that the type and level of statistical literacy

required can vary substantially in these roles. These requirements could also differ according

to the media product.

2) Why do you focus only on “SPIEGEL”. What would change, if we look at “BILD”,

“Spektrum der Wissenschaft” or international news papers or media? In my opinion, this

focus is much too narrow to achieve a good picture. I would find the results much more

exciting and valid if they were to expand them to include other (even very different) media.

Because with this very narrow view, no generalization is possible and I wonder what the

results mean at all.

Again, as pointed out above, we agree that extending the analysis to more newspapers,

which might differ in type, language, and other characteristics, would be very valuable. Given

limited access to full text and the substantial human resources required, we have to leave this

step to a follow-up project. However, we added the tabloid BILD for the same period as for

DER SPIEGEL following your suggestion. The differences found between the two newspapers

and the short time period do not allow for general conclusions but highlight the relevance of

the topic of our research. In the final section, we indicate our plans for future empirical

research, which should include further newspapers and a longer period of time.

Further issues:

“Statistical literacy”: Do you really mean statistical literacy or data literacy? These

concepts would have to be described in more detail, distinguished from one another and

consciously chosen. There is a wealth of literature available that explains the comparisons

between these two areas.

We agree that we must be more precise in using the term “statistical literacy”. In the text,

we explain that we use it in a rather broad sense, also comprising aspects of data literacy.

Abstract: “Statistical literacy” of whom? Of the reader, of the person who conducts

the study, presents the results, of the journalist . . . .? Please highlight in the abstract which

groups of people are involved.

As mentioned above, we tried to be more precise in pointing out which persons or groups

of people we address when referring to statistical literacy throughout the article, including

the abstract.

Introduction: “The first concerns the potential explanations for the discrepancies observed

among the figures – specifically, the variation between one-third, one-fourth, and

one-sixth” This comparison is very problematic (not only due to the two aspects you further

describe in the paper. The three different proportions are related to three different questions,

very sloppy formulated: “Proportion of men, who have resorted to physical violence“, “Proportion

of women, who get a victim of domestic violence”, “proportion of violence within a

partnership under victims of domestic violence”. Even though I understand that such a comparison

of very different percentage rates would be attractive, it is clearly misleading because

it does not look at the same proportion or the same question. The ability to distinguish

between such things would also be immensely important for readers of media reports.

Thank you for pointing out this potentially misleading presentation of the example. Given

that the result should only highlight that reports in the media might also be perceived quite

differently depending on the presented results and the way they are interpreted, we rephrased

this part to avoid the kind of misunderstanding you mentioned. We also added a remark on

the reporting of this study by DER SPIEGEL, while surprisingly, we could not find a mention

in BILD.

Figure 1: The arrow from media with “statistical literacy” does not fit. Here, as with

user, a person or group of people should also be named. And that is precisely what would

be interesting. WHO is involved in the chain here? Who all needs statistical literacy? And

what kind of statistical literacy in each case?

We agree that the actor media shown in the figure does not reflect well the different

actors involved. We modified the figure to make it clearer that we refer to these actors by

mentioning journalists, editors, and the publisher as examples of relevant actors. However,

we refrained from attempting to also describe their interaction in the figure, which would be

beyond the scope of the present article.

Introduction: When defining the concept of statistical literacy, please provide a citation.

We added more citations as proposed.

Allover the manuscript: Whenever you are speaking about “statistical literacy”, please

clarify the person or group of people, who needs statistical literacy, because in the whole

process of conducting the study (or the person or institute who commissions the study) to

reading the news, there are a lot of people who need statistical literacy.

As pointed out in the general statement above, we tried to be more precise about people

and institutions whenever referring to statistical literacy, keeping in mind that our model is

just a simplified abstract model of a quite complex network of actors.

Page 5 – line 201: “Until now, we have delineated the individual actors” No, you have

not; see comments above.

The phrase has been adjusted.

The article “Zhang H, Wang H. Information skills and literacy in investigative journalism

in the social media era. Journal of Information Science. 2022; p. 447–465. doi:10.1177/

01655515221094442.” you cited, has been retracted from the journal. I don’t know, what

might be an appropriate way of cite this article (or if it is even possible to cite this article

any more).

Thanks for pointing this out – the retraction took place after the citation was included

in an early draft of our paper. We decided to remove the reference and its content as the

provided evidence might be questionable.

Smaller Issues: - Page 2: “Section Economic Incentives , ” Please delete the space before

the comma. - Page 3: “In Section Insights into Empirics ,” the same

Thanks for pointing out the extra spaces. We deleted them.

2 Reviewer #2

The Impact of Statistical Literacy and Economic Incentives on the (Mis-)Use of Survey

Based Statistics in Media Reporting - A Framework

The paper provides a conceptual framework on how statistical literacy and economic

incentives influence presentation and understanding of survey data for media reports. The

authors clearly lay out the methodological shortcomings, principles of survey design and its

biases, and provide a conceptual framework that could perhaps serve as a foundation for

future empirical studies to better understand the several dynamics at play while evaluating

media reporting.

We thank the reviewer for summarizing the goals of our paper in such a precise and

concise way! This is exactly the purpose of the paper, while we leave a detailed empirical

analysis for future research.

I think this is extremely critical topic given the data intensive world we are all in. Also,

there is no doubt the conceptual framework provides a sense of direction for future research in

this area. I have a few comments and hope these reflections shall strengthen the manuscript.

We wholeheartedly agree that the topic is important given our focus on data as the basis

for decisions of all kind, e.g., policy, economics, elections. We tried to consider all your comments

in the revision and believe that these comments actually helped substantially improve

the presentation.

Also, the draft is well organized in terms of writing, and I enjoyed reading it!

We are indebted to the reviewer for this positive feedback on the structure and style of

writing.

1. I recommend the authors to clarify terms such as ”statistical literacy” early in the

manuscript to ensure consistency, and with appropriate definition that might be relevant

for journalism. For instance, is it just the problem of quantitative literacy? That is, would

numeracy literacy solve the problem? Or, it is more pervasive as it includes the relevance of

risk literacy as well.

We agree that the relevant terms such as “statistical literacy” should be defined early in

the manuscript and used consistently throughout. We made the relevant changes to achieve

this goal.

2. I encourage the manuscript to provide concrete examples of the ways in which news

reports systemically misinterpret, misrepresent, or misuse numerical data as part of the reporting

process, and probably ranging across diverse topics, such as healthcare, war, finance,

elections, poverty, etc. The point is that, for the world of Journalism, the issue of statistical

literacy is more pervasive and fundamental, than for a health-care expert or a finance expert,

where it could be more topical and functional. To elaborate, the issue of reporting relative

versus absolutive risks is very topical in Healthcare.

Further examples could better illustrate the relevance of different ways of reporting survey

results in the media. However, providing a systematic review of such issues is beyond the

scope of our paper. Nevertheless, we added some examples to highlight potential issues.

3. Though, the manuscript is purely conceptual and it does not include empirical evidence

to validate its proposed framework. Incorporating preliminary data or case studies would

strengthen its claims. Also, if it would be possible to analyze trends in media reporting using

a dataset of survey-based articles to identify patterns in statistical misrepresentation.

Given the conceptual nature of the paper, we just provided an initial empirical evidence

based on a single media outlet. Another reviewer also suggested extending the sample. Therefore,

we repeated our small-scale analysis for the tabloid BILD. However, studying trends in

media reporting in more detail would require a much larger dataset covering a longer time

period. Similarly, assessing the impact of the type of newspaper would require a larger crosssectional

dimension, i.e., more newspapers. This is planned for future empirical work subject

to obtaining the necessary resources, particularly access to full text, and for developing natural

language processing (NLP) methods to identify relevant documents and extract the pivotal

information.

4. While the paper discusses economic incentives driving media behavior, it does not delve

deeply into specific mechanisms (e.g., advertising revenue models or audience engagement

metrics). A more detailed analysis would enhance understanding. Also, if this relation could

be elaborated through case studies or through an empirical setting.

In fact, considering the specific economic incentives for different actors present in our

simplified model is another relevant extension starting from the framework. Unfortunately,

funding such an analysis would require access to data, which is currently not available. Thus,

we have to leave this issue for future research, but we mentioned its relevance in the concluding

section.

5. With regards to the issue of economic incentives, I am also pretty curious to know if

there are previous studies that clearly establish that statistical literacy would lead to more

accurate reporting and a more socially-responsible journalism? I ask the authors to think

about this question because a discussion on the trade-offs between the quality of surveybased

statistics and the economic profitability would be interesting to reflect. Would the

association btw statistical literacy and improved media reporting be so simple?! Although,

the authors sparingly mention about the trade-

---

## [Decision Letter · Decision Letter 1]

20 Jun 2025

PONE-D-25-03801R1The Impact of statistical Literacy and economic Incentives on the (Mis-)Use of Survey based Statistics in Media Reporting – A FrameworkPLOS ONE

Dear Dr. Bethäuser,

Thank you for submitting your manuscript to PLOS ONE. After careful consideration, we feel that it has merit but does not fully meet PLOS ONE’s publication criteria as it currently stands. Therefore, we invite you to submit a revised version of the manuscript that addresses the points raised during the review process.

We look forward to receiving your revised manuscript.

Kind regards,

Felix G. Rebitschek

Academic Editor

PLOS ONE

Additional Editor Comments:

Dear Authors,

Thanks for submitting your revision!

There are further comments on the revision by the reviewer and me.

Please take it into account:

Please check capitalisation of the title:

The Impact of statistical Literacy and economic Incentives on the (Mis-)Use of Survey based Statistics in Media Reporting – A Framework

The Abstract

It should include alternatives that are much more obvious for not reporting methodological background than economic incentives, e.g. the perception of journalists that the audience may be overstrained by that, or just space and time reasons, and collision with journalist principles like "pointing".

The method of the study should appear there.

Introduction

Please justify the focus on survey-based statistics clearly.

Please justify the focus on types of media [social media?, bloggers?].

What is the point about "surprising" and "negative" for your argumentation?

"In a limited sample of articles from the journal DER SPIEGEL and BILD..." Is this result or part of the argument for doing the study?

Please elaborate the argument if studies are commissioned more explicitly.

What is Plan International?

Consider taking the elaboration on the shortcomings of the Plan International Example as the starting point for outlining the motivation of the study - what are crucial methodological details that are omitted typically?

The third question leaves the shortcomings of the example, but aims at the procedural level.

Can you study the different levels at once?

"First, we describe the economic incentive structure for the whole pipeline from running a survey"

If this is the goal, the motivation need to be sharpened.

"Second, we also want to take into account the level of “statistical literacy” [of whom]?

The methodological section is crucial prior to your review

- of the theoretical and empirical literature on economic incentives for journalists and newspapers.

- of the role of statistical literacy for journalists

Why the focus on Germany?

Why the focus on those two media outlets?

Fig.1 should avoid correction marks included.

Fig.2: capitalisation, uncertainty information is missing, axis is not in the PDF, axis label is missing,

The results section should be clearly marked as such.

The discussion is missing and possibly mixed with implications.

Limitations of the current study are important.

Reviewers' comments:

Reviewer's Responses to Questions

**Comments to the Author**

1. If the authors have adequately addressed your comments raised in a previous round of review and you feel that this manuscript is now acceptable for publication, you may indicate that here to bypass the “Comments to the Author” section, enter your conflict of interest statement in the “Confidential to Editor” section, and submit your "Accept" recommendation.

Reviewer #1: (No Response)

Reviewer #2: All comments have been addressed

2. Is the manuscript technically sound, and do the data support the conclusions?

Reviewer #1: Yes

Reviewer #2: Yes

3. Has the statistical analysis been performed appropriately and rigorously? 

Reviewer #1: Yes

Reviewer #2: Yes

4. Have the authors made all data underlying the findings in their manuscript fully available?

Reviewer #1: Yes

Reviewer #2: Yes

5. Is the manuscript presented in an intelligible fashion and written in standard English?

Reviewer #1: Yes

Reviewer #2: Yes

6. Review Comments to the Author

Reviewer #1: I am reviewing this manuscript for the second time. As I have already mentioned before: The topic and the research question addressed in the manuscript is of enormous relevance in a world in which more and more statistics are disseminated through the media and in which there is a great demand for statistics that are both good and well reported. I am very pleased that the authors have also provided a little empirical insight into articles in BILD. This extension is quite exciting and an enhancement of the article.

Overall, I think that the revision of the article has been successful and I only have minor aspects that need to be revised, which I will list in the following:

Minor issues:

1) I believe that the authors are still not sufficiently aware of which and how many actors are involved in certain media articles and other media products and that these require a very different kind of statistical literacy.

2) You provide a definition of “statistical literacy” very late in the article. Furthermore, there is the headline “Impact of statistical literacy” in where you first describe scientific literacy, which is quite confusing for readers. It would be better to refer to this section already when you first mention statistical literacy at page 3, line 92.

Furthermore, the definition and explanation of statistical literacy is still weak. When reading your article, I sometimes thought about the quote from H.G. Wells, who said “Statistical thinking will one day be as necessary for efficient citizenship as the ability to read and write!”

3) Figure 1 contains markings from the grammar check. Please delete them.

4) Line 202 “still contains markings from the grammar check. Please delete them.” In sentences like this, it seems as if you have thought through all the groups of people in detail. But I think that's missing from the paper. If you do not fulfill this (as you write in the answer - which is also okay), please weaken these sentences accordingly.

5) Line 404: “Statistical literacy may be essential for journalists when selecting high-quality survey results to report on, …” This is a very narrow few of statistical literacy for journalists. Statistical literacy is not only necessary for evaluating the quality of the studies.

6) Line 505: 21. 89% Please delete the blank space.

7) Line 581: Please add an example for a statistical key word you mean here.

8) Figure 2: The percentage sign is not placed very well. Furthermore, it would be great to provide the full area to 100% (instead of 70%).

9) Line 626: It would be great to insert the 9 out of 144 also in Figure 2.

10) Line 628: Here you are speaking about red bars, whereas in Figure 2 you are speaking about orange bars.

11) Line 650: Is the study really about a theoretical framework for survey reporting in the media? I think what you write in the footnote of Figure 1 matches better.

12) Line 651 “we have systematically analyzed the role of different actors”: No you have not! Please delete “systematically”

13) Sometimes you speak about a pilot study, a small-scale and exploratory empirical pilot study, an exploratory small-scale pilot study. Please standardize throughout the article.

Reviewer #2: I am sufficiently satisfied with the responses. My only comment is to send the draft for copy editing.

7. PLOS authors have the option to publish the peer review history of their article (what does this mean?). If published, this will include your full peer review and any attached files.

Reviewer #1: No

Reviewer #2: **Yes: **Kavitha Ranganathan

---

## [Author Response · Author response to Decision Letter 2]

28 Jul 2025

Response to Editor and Reviewers for Revised Article

“The Impact of Statistical Literacy and Economic Incentives on the (Mis-)Use of Survey based Statistics in Media Reporting — A Framework”

July 27, 2025

We are indebted to the reviewers for reading again the revised version of our paper and the further comments received from one reviewer and the editor. We are pleased that by our first revision we were able to address all points originally raised by reviewer #2 and also most of those indicated by reviewer #1. In the following, we will describe how the remaining minor comments of reviewer #1 and the helpful guidelines provided by the editor were addressed.

The original comment is repeated first, followed by our response in italics.

1 Reviewer #1

I am reviewing this manuscript for the second time. As I have already mentioned before: The topic and the research question addressed in the manuscript is of enormous relevance in a world in which more and more statistics are disseminated through the media and in which there is a great demand for statistics that are both good and well reported. I am very pleased that the authors have also provided a little empirical insight into articles in BILD. This extension is quite exciting and an enhancement of the article. Overall, I think that the revision of the article has been successful and I only have minor aspects that need to be revised, which I will list in the following:

We are glad that the reviewer appreciates our added analysis for BILD, which provides first insights into the heterogeneity that could be expected across different types of news outlets. It is good news for us that we could address all the major issues from her/his first report and did our best to deal with the remaining minor issues as described below.

Minor issues: 1) I believe that the authors are still not sufficiently aware of which and how many actors are involved in certain media articles and other media products and that these require a very different kind of statistical literacy.

The reviewer comes back to his argument that our framework does not cover all relevant actors and that the role of different actors might require different kinds of statistical literacy. We agree on these limitations and – also in accordance with comment 4) and a related comment by the editor tried to make them more transparent throughout the paper.

2) You provide a definition of “statistical literacy” very late in the article. Furthermore, there is the headline “Impact of statistical literacy” in where you first describe scientific literacy, which is quite confusing for readers. It would be better to refer to this section already when you first mention statistical literacy at page 3, line 92. Furthermore, the definition and explanation of statistical literacy is still weak. When reading your article, I sometimes thought about the quote from H.G. Wells, who said “Statistical thinking will one day be as necessary for efficient citizenship as the ability to read and write!”

We are indebted to the reviewer for this comment. We see the point that the first discussion of statistical literacy on page 3, line 92 (in the version with track changes) is not comprehensive enough. Thus, we follow the suggestion of adding a reference to the section 'Impact of Statistical Literacy’, where a more comprehensive discussion is provided. In that section, we deliberately introduce both scientific and statistical literacy in order to distinguish between the two concepts, as we explicitly state. That said, we fully agree with the spirit of the H.G. Wells quote and appreciate its relevance.

3) Figure 1 contains markings from the grammar check. Please delete them.

Thanks for pointing out this shortcoming of the previous version, which has been corrected.

4) Line 202 “still contains markings from the grammar check. Please delete them.” In sentences like this, it seems as if you have thought through all the groups of people in detail. But I think that’s missing from the paper. If you do not fulfill this (as you write in the answer - which is also okay), please weaken these sentences accordingly.

The markups you mention show up in the revised manuscript with track changes, but not in the manuscript itself. They are shown in the version with track changes to highlight new parts and removed parts in addition to the different colors. Your comment on the groups of people is related to comment 1) and similar to a statement made by the editor. Therefore, we checked all sentences regarding the groups of people considered in the framework and made adjustments to avoid the impression that the article provides a complete picture of all potentially relevant groups of people and a detailed description of all factors driving their decisions.

5) Line 404: “Statistical literacy may be essential for journalists when selecting highquality survey results to report on, . . . ” This is a very narrow view of statistical literacy for journalists. Statistical literacy is not only necessary for evaluating the quality of the studies.

We fully agree and have expanded the relevant section to highlight additional advantages statistical literacy offers journalists beyond evaluating study quality.

6) Line 505: 21. 89% Please delete the blank space.

Actually in line 555 of the revised version with track changes: Done.

7) Line 581: Please add an example for a statistical key word you mean here.

As described in the text, we identified various statistical keywords such as ’representative’, ’error rate’, ’sampling procedure’, and others. For better clarity, we have now also included them in the description of the figure categories, as you suggested.

8) Figure 2: The percentage sign is not placed very well. Furthermore, it would be great to provide the full area to 100% (instead of 70%).

We changed the scale to 0 – 100%.

9) Line 626: It would be great to insert the 9 out of 144 also in Figure 2.

One way of including this information in Figure 2 would be to add an additional category. This would not change anything to the blue bars, but reduce the height all of remaining organge bars (dividing by 144 instead of by 135). However, since our focus is on the reporting on surveys, this might be misleading since the 9 articles do not refer to any specific survey. Therefore, we prefer to exclude these articles from the figure. However, the qualitative result would not be affected.

10) Line 628: Here you are speaking about red bars, whereas in Figure 2 you are speaking about orange bars.

Thanks for pointing out this error referring to an earlier version of the figure with red bars. We changed to “orange bars”.

11) Line 650: Is the study really about a theoretical framework for survey reporting in the media? I think what you write in the footnote of Figure 1 matches better.

We agree that the presentation of the small-scale empirical pilot study might suggest a more narrow perspective. However, the focus of our paper is on a theoretical framework for survey reporting in the media. We clarified this throughout the text.

12) Line 651 “we have systematically analyzed the role of different actors”: No you have not! Please delete “systematically”

In accordance with our reply to your points 1) and 4) we agree that the statement might be too strong. Therefore, we removed the word ”systematically” as suggested.

13) Sometimes you speak about a pilot study, a small-scale and exploratory empirical pilot study, an exploratory small-scale pilot study. Please standardize throughout the article.

Thank you for pointing out the inconsistency. We have adjusted the formulation accordingly.

2 Editor

Please check capitalization of the title: The Impact of statistical Literacy and economic

Incentives on the (Mis-)Use of Survey based Statistics in Media Reporting – A Framework

Done.

The Abstract

It should include alternatives that are much more obvious for not reporting methodological background than economic incentives, e.g. the perception of journalists that the audience may be overstrained by that, or just space and time reasons, and collision with journalist principles like ”pointing”. The method of the study should appear there.

We have revised the abstract to include the additional reasons you suggested for omitting methodological details, and we have also referenced our small-scale empirical pilot study of BILD and DER SPIEGEL to support the relevance of our framework.

Introduction Please justify the focus on survey-based statistics clearly. Please justify the focus on types of media [social media?, bloggers?].

Thanks for pointing out that these aspects need justification. We clarified the focus onsurvey statistics and included an argument why we focus on mass media.

What is the point about ”surprising” and ”negative” for your argumentation?

Thank you for pointing this out. We agree that the notions of “surprising” and “negative” findings are not central to the current analysis. Although such aspects might be relevant in future empirical studies exploring how the framing or perceived news value of survey results influences media uptake, they are not further discussed in this paper. For this reason, we have removed the respective part of the paragraph to maintain a clear focus.

”In a limited sample of articles from the journal DER SPIEGEL and BILD...” Is this result or part of the argument for doing the study? Please elaborate the argument if studies are commissioned more explicitly.

Thank you for this helpful question. We have clarified our argument in the manuscript as to why it is relevant to examine the share of commissioned studies. We do not claim that commissioned studies are more likely to be covered in the media, as we do not have information on surveys that were not reported. However, we observe that a considerable share of the surveys in our sample were commissioned. Although we cannot draw causal conclusions from this, we have decided to keep this observation as it may be relevant for future research, particularly regarding the topics and contexts in which surveys are commissioned. We added a sentence in the introduction and mention the issue when discussing limitations to highlight that this finding of our empirical analysis deserves attention in future research.

What is Plan International? Consider taking the elaboration on the shortcomings of the Plan International Example as the starting point for outlining the motivation of the study - what are crucial methodological details that are omitted typically?

Thank you for this helpful comment. We have added a brief explanation of Plan International, including a link to the NGO in the manuscript to ensure clarity for all readers. The findings of the Plan International study serve primarily to motivate our research, as they clearly illustrate that there are differences in the way surveys are reported in the media. However, this example is not particularly well suited to discuss methodological shortcomings in detail, beyond the issues we already mentioned, such as representativeness and selection bias. For this reason, we refer to it mainly in the context of media attention rather than methodological critique.

The third question leaves the shortcomings of the example, but aims at the procedurallevel. Can you study the different levels at once?

We are not entirely sure what your comment refers to. Just to clarify, the third research question was: “Third, how were these results presented to readers and how much attention did they receive?” We do return to the example of the Plan International study, as it raised the question of why this particular survey received extensive media coverage, while official statistics on the same topic did not. Our theoretical framework highlights some factors that may influence whether and how surveys are reported in the media, such as funding sources or the speed of result availability. Although we cannot empirically determine the exact reasons why certain surveys receive more attention than others, we can observe and analyze the fact that they were reported. We acknowledge that this limits our ability to study all levels simultaneously, but we believe that our approach offers valuable information on the dynamics of media coverage and survey visibility.

”First, we describe the economic incentive structure for the whole pipeline from running a survey” If this is the goal, the motivation need to be sharpened. ”Second, we also want to take into account the level of “statistical literacy” [of whom]?

We changed the description of the framework as one that defines the general actors involved in the production of mass media with a focus on the incentives in reporting the results of surveys by mass media. We specified that statistical literacy is relevant to media and users as actors.

The methodological section is crucial prior to your review - of the theoretical and empirical literature on economic incentives for journalists and newspapers. - of the role of

statistical literacy for journalists

For the report on the empirical study, we introduced the section ’Methods’ that describes the decision to use both outlets, the key methods of text selection, data cleaning, and analysis. Consequently, the results of the study are presented in the ’Results’ section.

Why the focus on Germany? Why the focus on those two media outlets?

We focus on Germany to examine the behavior of the media in a high-income democracy with strong media institutions. BILD and DER SPIEGEL were selected due to their wide reach and contrasting editorial styles, which allow comparative insight into survey reporting practices. We discuss this in detail in ’Methods’ (p. 10ff), where we outline the relevance of BILD and SPIEGEL for our research focus. As noted in the final paragraph of the paper, we would very much like to extend our analysis to additional media outlets and countries in the future, once this can be done using automated methods rather than manual coding.

Fig.1 should avoid correction marks included.

Done following also the recommendation by Reviewer #1.

Fig.2: capitalization, uncertainty information is missing, axis is not in the PDF, axis

label is missing,

We assume that ”capitalization” refers to the labels of the two newspapers. In fact, both newspapers use capitalization as part of the brand. Therefore, we also use it both in the text and in Figure 2, whenever referring to them. We made the following changes to the figure: added error bars indicating +/- one standard

deviation for the shares, modified the y-axis to 0 - 100 % as suggested by reviewer #1 and added axis arrows and axis labels.

The results section should be clearly marked as such. The discussion is missing and possibly mixed with implications. Limitations of the current study are important.

We included the ’Results’ section. We renamed the ’Implications’ into ’Discussion’, inwhich the limitations of the current study are stated more clearly.

---

## [Editor Report · Decision Letter 2]

11 Aug 2025

PONE-D-25-03801R2The impact of statistical literacy and economic incentives on the (mis-)use of survey based statistics in media reporting – A frameworkPLOS ONE

Dear Dr. Bethäuser,

Thank you for submitting your manuscript to PLOS ONE. After careful consideration, we feel that it has merit but does not fully meet PLOS ONE’s publication criteria as it currently stands. Therefore, we invite you to submit a revised version of the manuscript that addresses the points raised during the review process.

We look forward to receiving your revised manuscript.

Kind regards,

Felix G. Rebitschek

Academic Editor

PLOS ONE

Journal Requirements:

Additional Editor Comments (if provided):

Dear Authors,

Thanks for addressing the manifold comments!

The following minor modifications are still important:

Method and Results have to be written in past tense (see the abstract).

Rethink the title please, because it should be much closer to the actual insight that your method allowed to investigate.

Your justification for choosing “mass media“ evokes the question for the underlying definition of mass media – for instance you indicate they are considered “high-quality media“ – do you understand mass media as equaling the “leading media“, the traditional media

Introduction ”In a limited sample of articles from the journal DER SPIEGEL and BILD...” – you should refrain from placing results in the introduction, where you motivate the study producing the results

Capitalization should be consistent across all terms in all figures (e.g., perfect in Figure 1).

---

## [Author Response · Author response to Decision Letter 3]

4 Sep 2025

Response to Editor for 3rd Revision of Article "How survey results are reported in the media: A framework on selection mechanisms and a pilot study on reporting practices", formerly titled “The Impact of Statistical Literacy and Economic Incentives on the (Mis-)Use of Survey based Statistics in Media Reporting — A Framework”

August 26, 2025

We appreciate that the editor acknowledges that our second revision could deal with the comments from one reviewer and himself in an adequate way. In the present final revision, we address the following minor modifications recommended by the editor:

The original recommendation is repeated first, followed by our response in italics.

– Method and Results have to be written in past tense (see the abstract).

We changed the description of Methods and Results to past tense throughout the paper

including the relevant sentences in the abstract.

– Rethink the title please, because it should be much closer to the actual insight that your method allowed to investigate.

Based on an earlier comment from one of the reviewers, we already thought about alternative titles that fit better the actual content of the article. We propose the following modified title:

”How survey results are reported in the media: A framework on selection mechanisms and a pilot study on reporting practices” or, if you prefer a more concise title, “Survey reporting in the media: A framework and a pilot study”

– Your justification for choosing ”mass media“ evokes the question for the underlying definition of mass media – for instance you indicate they are considered ” high-quality media“ – do you understand mass media as equaling the leading media“, the traditional media.

In this paper, we use the term “mass media” to refer to traditional, widely circulated news outlets with large audiences and significant agenda-setting power. This includes both high-quality investigative journalism (e.g., DER SPIEGEL) and more populist formats (e.g., BILD). While these outlets differ in tone and editorial standards, they share the characteristic of reaching mass audiences through institutionalized journalistic practices.

We changed the wording from ”the mass media is recognized as high-quality media, relevant for the formation of opinions of citizens” to ”the mass media are recognized as widely circulated news outlets with large audiences and significant agenda-setting power, relevant for the formation of opinions of citizens”.

– Introduction ”In a limited sample of articles from the journal DER SPIEGEL and BILD...” – you should refrain from placing results in the introduction, where you motivate the study producing the results

As suggested we removed the part presenting results in the introduction.

– Capitalization should be consistent across all terms in all figures (e.g., perfect in Figure 1).

We adjusted capitalization to use capital letters only for the first term in each box, for each axis etc.

---

## [Editor Report · Decision Letter 3]

9 Sep 2025

How survey results are reported in the media: A framework on selection mechanisms and a pilot study on reporting practices

PONE-D-25-03801R3

Dear Dr. Bethäuser,

We’re pleased to inform you that your manuscript has been judged scientifically suitable for publication and will be formally accepted for publication once it meets all outstanding technical requirements.

Kind regards,

Felix G. Rebitschek

Academic Editor

PLOS ONE

Additional Editor Comments (optional):

Please keep the more extensive of the new titles.
---

## [Editor Report · Acceptance letter]

PONE-D-25-03801R3

PLOS ONE

Dear Dr. Bethäuser,

I'm pleased to inform you that your manuscript has been deemed suitable for publication in PLOS ONE. Congratulations! Your manuscript is now being handed over to our production team.

Kind regards,

on behalf of

Dr. Felix G. Rebitschek

Academic Editor

PLOS ONE